# Polyhydroxybutyrate Rice Hull and Torrefied Rice Hull Biocomposites

**DOI:** 10.3390/polym14183882

**Published:** 2022-09-17

**Authors:** Zach McCaffrey, Andrew Cal, Lennard Torres, Bor-Sen Chiou, Delilah Wood, Tina Williams, William Orts

**Affiliations:** 1Agricultural Research Service, US Department of Agriculture 800 Buchanan Street, Albany, CA 94710, USA; 2Mango Materials, 490 Lake Park Ave, Oakland, CA 94610, USA

**Keywords:** particle matrix composite, polyhydroxybutyrate (PHB), plastic composite, torrefaction, biodegradable, rice hulls

## Abstract

Raw and torrefied rice hulls (RRH and TRH) were incorporated into polyhydroxybutyrate (PHB) as fillers using extrusion and injection molding to produce biomass-polymer composites. Filler and composite materials were characterized by particle size analysis, thermomechanical analysis, thermogravimetric analysis, differential scanning calorimetry, FTIR analysis, CHNSO analysis, and mechanical testing. Heat distortion temperature of the RRH composites were 16–22 °C higher than TRH composites. The RRH composite samples showed a 50–60% increase in flexural modulus and 5% increase in stress at yield compared to PHB, while TRH composite samples showed nearly equal flexural modulus and a 24% decrease in stress at yield. The improved mechanical properties of the RRH composites in comparison to TRH composites were due to better particle-matrix adhesion. FTIR analysis showed RRH particles contained more surface functional groups containing oxygen than TRH particles, indicating that RRHs should be more compatible with the polar PHB plastic. SEM images showed space between filler and plastic in TRH composites and better wetted filler particles in the RRH composites.

## 1. Introduction

Plastic production has increased at an enormous rate since the 1950s, but much of the plastic remains in the environment decades after its usefulness [1]. Plastics often find their way into our water systems, including oceans and rivers [2,3], are disposed of on the ground [4], are eaten by fish [5] and many other animals [6], are mixed with air [7], and are found in human bodies [8]. In 2020, over 360 million metric tons of plastic were produced worldwide, with over 40% manufactured for single use products [9], and 50% of it becoming trash in less than a year [10]. About 9% of all plastic ever made has been recycled, 12% incinerated, and 79% landfilled or accumulating in the environment [10]. To address this problem, increased efforts are being made to develop biodegradable bio-based polymers from renewable resources to reduce the accumulation of persistent plastics [11].

One of the most promising biodegradable and renewable bio-based plastics are polyhydroxyalkanoates (PHAs) [12,13], a family of biopolyesters that can be produced from various strains of bacteria [14], archaea [15], and algae [16]. PHAs can be homo- and copolymers (e.g., P(3HB-co-3HHx), P(3HB-co-4HB)) and have properties [17] that can mimic petroleum-based plastics, such as polyethylene and polypropylene. Polyhydroxybutyrate (PHB) is a bioprocessed polyester belonging to the PHA family and is characterized by having a methyl functional group (CH_3_) and an ester linkage group (−COOR), which result in the material’s hydrophobic and thermoplastic characteristics [17]. Many bacterial strains produce PHB as a carbon reserve and, industrially, PHB is produced by bacterial cultivation [18]. PHB has several advantages over petroleum-based polymers such as polyethylene (PE) and polypropylene (PP). For example, PHB has superior barrier permeability, is more rigid and less flexible, and is biodegradable in soils, fresh water, and under composting conditions [17].

However, P(3HB), the simplest, most extensively studied type of PHA (Briassoulis et al., 2021), is highly crystalline (up to 70% crystallinity) [19] and not fully compatible with some production processes, resulting in higher production costs, greater complexity in manufacturing, and greater complexity in downstream processing [20]. One solution to this problem is blending P(3HB) with fillers or other compounds to improve the composite’s physical properties and reduce the cost of manufacturing [21].

Rice hulls are an underutilized byproduct of the rice industry and have potential as a filler in plastics due to their material properties, abundance, and low cost. Rice is a primary source of food for over half the world’s population and 515 million tons on a milled basis are projected to be produced in 2022–2023 [22]. Rice hulls make up about 20% of harvested rice, with the remainder being the endosperm (72%), which is the commonly eaten part, and rice bran (8%). Rice hulls have a density of 80–125 kg/m^3^; consist of 60–65% volatile matter, 10–15% fixed carbon, and 17–23% ash [23] with the ash content rich in silica [24]. Rice hulls have been used as an additive in many materials, including animal feed [25], refractory brick [26], insulation [27], flame retardant materials [28], rubber [29], and plastic composites [21] among others.

The combination of renewable-based biodegradable polymer together with a natural fiber filler creates a biocomposite that is a promising alternative to petroleum-based plastics with benefits that include: (1) reduction in the dependence on petroleum-based materials and increased use of renewable sources in manufacturing [17]; (2) smaller end-of-life environmental impact, since there is no need to landfill or incinerate them [30]; and (3) increased value of agricultural byproducts used as fillers, which increases farm revenues, reduces the overall impact of the food production cycle, and promotes a circular economy [31]. It is noteworthy that rice hulls are generally amassed by rice processors during rice-milling, with limited value-added outlets for their use. Adding hulls to plastics has the potential to improve the economics of rice processing by providing a consistent outlet for hulls, avoiding landfilling.

Torrefaction of biomass consists of thermally heating to temperatures between 200 and 300 °C in an inert atmosphere [32]. Torrefaction of biomass results in a material with a darker color, greater friability due to decomposition of hemicelluloses and partial decomposition of the lignin and cellulose components [33], higher hydrophobicity, higher calorific value, and reduced moisture absorption compared to the original raw biomass [34]. Increased hydrophobicity is desirable for polymer reinforcement applications and leads to improved filler-matrix adhesion [35]. Previous research on polypropylene-polyethylene composites with almond shells showed that torrefaction of biomass improved grindability, which resulted in smaller particle sizes and improved heat deflection temperatures. This was due to good adhesion between the filler and matrix, with no need for compatibilizers [36]. To the best of our knowledge, there have been no previous studies on using torrefied biomass as fillers in PHB composites, although there has been one study on torrefied fillers in PHBV composites [37]. In addition, there have been only a couple of studies that involved incorporating biochar as fillers in PHB composites and those studies did not focus on mechanical properties of the PHB composites [38,39].

In the present work, the effects of raw and torrefied rice hulls on PHB composite properties were investigated. Two different filler particle sizes were examined in the study. The composites’ flexural and tensile properties were determined by strain-gauge testing, heat deflection temperature by thermo-mechanical analysis (TMA), crystallinity by differential scanning calorimetry (DSC), adhesion by scanning electron microscopy (SEM), and matrix composition by Fourier transform infrared (FTIR) spectroscopy. The material properties of the PHB composites were analyzed to develop materials suitable for consumer plastic applications.

## 2. Methods and Materials

### 2.1. Materials

PHB was obtained from TainAn Biopolymers (Ningbo, China). The material was sold as PHBV, but the HV content was less than 1% by gas chromatography and had a melting point of 175 °C. PHB was blended with 10% tributylacetylcitrate (Chempoint, Bellevue, WA, USA) and 0.1% Luperox TAEC (Arkema, Colombes, France) during extrusion. Rice hulls were obtained from French Camp Grain Elevators (French Camp, CA, USA) and were received with moisture content of 8.0%.

### 2.2. Torrefaction

The rice hulls were torrefied using a high temperature muffle furnace (Thermo Fisher, Waltham, MA, USA). Three-hundred grams of as-received raw rice hulls were placed within a steel vessel in the furnace. Two additional thermocouples were added inside the furnace to monitor temperatures during the experiment: (1) inside the steel vessel, within the pile of rice hulls, and (2) outside the steel vessel, close to the furnace floor. Next, the steel vessel was purged with nitrogen gas (2 L min^−1^) for 30 min prior to heating and continued throughout the experiment to displace the initial air and maintain an inert gas environment. The furnace was heated to a set temperature of 275 °C for 4 h. Then, the torrefied biomass was allowed to cool to room temperature, after which the nitrogen flow was turned off.

### 2.3. Milling and Particle Size Analysis

Raw and torrefied rice hulls were first ground using a Wiley knife mill with a 5 mm screen. The hulls were ground further to their final size using one of two mills: IKA knife mill using a 1 mm screen (coarse mill), and Union Process attritor (Cuyahoga Falls, OH, USA) with ¼” alumina milling balls (fine mill), for 30 min. This resulted in rice hulls produced under four conditions: raw-coarse mill, raw-fine-mill, torrefied-coarse mill, and torrefied-fine mill. The particle size of each sample was measured using a Horiba LA-960 (Piscataway, NJ, USA) dynamic particle size analyzer at a refractive index of 1.47.

### 2.4. CHNSO Analysis

A CHNSO analyzer (Vario el Cube, Elementar, Langenselbold, Germany) was used to analyze biomass samples for carbon (C), hydrogen (H), nitrogen (N), oxygen (O), and sulfur (S). Each sample was analyzed using CHNS mode and O mode of the instrument separately and in triplicate. The instrument was calibrated using benzoic acid and sulfanilic acid standards.

### 2.5. Differential Scanning Calorimeter (DSC)

Percent crystallinity of the composite samples was measured using a differential scanning calorimeter (Perkin Elmer DSC 8000, Waltham, MA, USA).

Test samples of the composite (8–12 mg) were placed in crimped aluminum pans for analysis. Sample and reference pans were simultaneously heated from 30 °C to 200 °C at a rate of 10 °C min^−1^ and purged with nitrogen gas. The heats of melting (Δ*H_m_*) were determined by integrating the DSC curves. Percent crystallinity (%*X*) was calculated by:(1)$$\%X=\frac{\Delta H_{m}}{\Delta H_{m}^{o}}*\frac{1}{\left(1-y\right)}$$
where $\Delta H_{m}^{o}$ is a reference value for heat of melting of a 100% crystalline sample, and y is the particle loading fraction. The reference value for heat of melting is 146 J g^−1^ for PHB [40].

### 2.6. Thermogravimetric Analysis (TGA)

Thermogravimetric analyses were performed using a Mettler Toledo TGA-DSC 3+ (Mettler-Toledo, LLC, Columbus, OH, USA) instrument. Samples were heated from 25 °C to 900 °C at 10 °C min^−1^ with nitrogen flow of 30 mL min^−1^. Alumina crucibles with 900 µL volume were used.

### 2.7. Extrusion and Injection Molding

A co-rotating twin-screw extruder (Leistritz Micro 18, Somerville, NJ, USA) was used to extrude PHB composite materials into strands. The twin-screw extruder had six heated zones held at temperatures of 150, 160, 170, 175, 180, and 180 °C from feed to die. Each of the two screws had a diameter of 18 mm. The extruder’s barrel had a length to diameter ratio of 30-to-1. All composite samples contained 10 wt% filler. An injection molder (Boy Machines 15 S, Hauppauge, NY, USA) was used to manufacture testing strips from the pelletized strands for Instron mechanical analysis. Each of the three temperature zones of the injection molder was set at 175 °C.

### 2.8. Mechanical Testing

Flexural and tensile tests of the biocomposites were performed using an Instron (Canton, MA, USA) 5500R universal testing machine. The ASTM D7264 method (Procedure A) was used for 3-point flexural testing and the ASTM D882-02 method was used for tensile tests. The three-point flexural test was performed by placing a rectangular specimen (length of 63.5 mm, width of 12 mm, and width of 1.5 mm) on two supports and applying a load (1-kN load cell, 25.4 mm min^−1^ extension rate) at the midpoint between the supports. For tensile testing, a rectangular specimen (150 mm × 5 mm × 0.5 mm) was placed between two pneumatic grips that were 100 mm apart from each other. Prior to mechanical testing, samples were conditioned at room temperature (~25 °C) and relative humidity (near 50%) for 24 h. Sample measurements were performed in at least triplicate.

### 2.9. Fourier Transform Infrared (FTIR) Spectroscopy

IR spectra of each sample was measured using an FTIR analyzer in absorbance mode (Thermo Fisher Scientific Nicolet iS10, Waltham, MA, USA) by averaging 16 scans between 4000 and 500 cm^−1^ with 4 cm^−1^ resolution.

### 2.10. Thermomechanical Analysis (TMA)

Heat distortion temperature (HDT) was determined using a thermo-mechanical analyzer (TMA 2940, TA Instruments, New Castle, DE, USA) and following the ASTM E2092-09 Method A. Test bars measuring 10 mm × 5 mm × 1.5 mm were cut from the injection molded bars. The test bars were heated from 25 °C to 200 °C at 2 °C min^−1^. A strain rate of 2 mm m^−1^ was used to determine heat distortion temperature. Endpoint deflection (*D*) in µm was determined by the following equation:(2)$$D=\frac{r\;L^{2}}{6\;d}$$
where *r* was the strain (mm m^−1^), *d* was sample thickness (mm), and *L* was distance between the two supports (5 mm). HDT was reported as the average of duplicate measurements.

### 2.11. Scanning Electron Microscope

Biocomposite samples were cryo-fractured in liquid nitrogen to reveal cross section surfaces and mounted onto aluminum specimen stubs using double coated carbon adhesive tabs (Ted Pella, Inc, Redding, CA, USA). The mounted samples were sputter coated with gold-palladium in a Denton Desk II sputter coating unit (Moorestown, NJ, USA) for 45 s using a discharge current of 20–30 mA and vacuum chamber pressure of 13 Pa (100 mTorr). Samples were viewed and photographed in a JEOL JSM 7900F field emission scanning electron microscope at 2.0 kV (Tokyo, Japan). Images were recorded at 2560 × 2048-pixel resolution.

## 3. Results and Discussion

### 3.1. Milling and Particle Size Analysis

Figure 1 shows the particle size distributions for fine and coarse raw rice hulls (RRH) as well as fine and coarse torrefied rice hulls (TRH), both by frequency distribution and cumulative percentage. Mean particle sizes for fine TRH, fine RRH, coarse TRH, and coarse RRH were 22 µm, 36 µm, 240 µm, and 445 µm, respectively. Coarse samples were milled with the IKA knife mill and fine samples were milled with the attritor. Milled torrefied samples resulted in smaller particle size distribution compared to milled raw samples, using the same milling equipment and operated under the same conditions.

The smaller particle size distribution of torrefied biomass samples was due to their easier grindability compared to raw feedstock. The heat treatment process of torrefaction causes significant transformation in the chemical composition of the biomass through dehydration, hydrolysis, oxidation, decarboxylation, and transglycolsylation [41]. Hemicellulose is the most reactive of the three lignocellulosic components due to its relatively low molecular weight and lower degradation temperature (160–250 °C) than both cellulose and lignin [42]. Mechanical properties of the biomass were altered as the decomposition of long polymers caused a decrease in biomass elasticity. The biomass became more friable and brittle [43], which resulted in increased grindability. The non-singular peak of RRH-fine sample shows that intermediate-sized particles were generated during the milling process using the attritor. [44] similarly found that intermediate peaks are formed during jet milling sand. The RRH-fine sample likely would have achieved a more singular peak distribution similar to the TRH-fine sample with greater milling time or milling energy.

### 3.2. CHNSO Analysis

Table 1 presents the elemental compositions of torrefied and raw rice hulls. Torrefaction resulted in rice hulls with increased C, N, and ash contents and decreased H, S, and O contents. Ash content of raw rice hulls was 18.73%, which is very high compared to most biomass types [45]. Raw rice hulls had H/C and O/C atomic ratios of 1.5 and 0.7, respectively, whereas torrefied rice hulls had H/C and O/C ratios of 1.2 and 0.3, respectively. For reference, sub-bituminous coal has H/C and O/C ratios of between 0.5 and 0.7 and between 0.2 and 0.3, respectively [46]. Most significantly, the concentration of ash content increased, and the concentration of oxygen decreased in the torrefied samples. These results are consistent with previous studies reporting elemental analysis of raw and torrefied rice hulls [47,48].

### 3.3. Differential Scanning Calorimeter (DSC)

The effects of particle size and heat treatment on crystallinity of the composites are shown in Table 2. PHB and all four composites had crystallinity values between 54.2 and 55.0 percent, which indicated that compounding with fillers had a negligible effect on the crystallinity of the composites. The results for the RRH samples were consistent with those found in PHB composites containing rice hulls treated with NaOH and H_2_O_2_ [49]. In that study, the treated rice hulls had little effect on the crystallinity of PHB in the composites.

### 3.4. Thermogravimetric Analysis

The main components of lignocellulosic biomass are hemicellulose, cellulose, and lignin. In previous research, the thermogravimetric (TG) and derivative TG (DTG) curves of cellulose (Avicel), hemicellulose (Xylan), and lignin were determined over the temperature range of 40–900 °C [50] to demonstrate typical decomposition temperatures for the three components. The results showed hemicellulose degradation began at 230 °C and had two peaks, one due to acetyl fragmentation reactions at 250 °C [51] and the second due to depolymerization reactions at 290 °C [52]. Cellulose degraded mainly between 300 and 400 °C and lignin had a relatively slow degradation rate between 250 and 900 °C.

The results of the TG and DTG analyses for raw and torrefied rice hull samples are shown in Figure 2. As the sample was heated, volatile compounds were released and, correspondingly, solid sample mass decreased. DTG curves were calculated by taking the derivative of the TG curve with respect to temperature. DTG peaks indicated the occurrence of the greatest mass reductions. [53] reported water evaporation occurred between 50 and 150 °C, hemicellulose degraded between 200 and 350 °C, cellulose degraded between 250 and 400 °C, and lignin degraded between 150 and 1000 °C. The DTG analyses of rice hulls confirmed this general behavior, with peaks at 120 °C for moisture, 290 °C for hemicellulose, and 330 °C for cellulose, followed by tapering mass loss from 400 to 900 °C. The remaining components at the end of the analyses were predominantly fixed carbon and ash, similar to products from proximate analyses prior to adding air or oxygen. TG curves for RHH and TRH were nearly identical up to 250 °C, after which RHH had faster mass loss due to hemicellulose degradation. The DTG curves confirmed TRH had almost no degradation (other than the moisture peak) until close to 300 °C, indicating volatiles below 300 °C had already been removed during torrefaction.

### 3.5. Mechanical Testing

Figure 3 shows representative tensile stress–strain curves for PHB and PHB-rice hull composites. Tensile modulus, tensile strength, and elongation were calculated from the stress–strain curves. Mechanical properties were statistically analyzed using ANOVA and Tukey’s test (α = 0.05) to determine whether there were significant differences between the plastic composites. Means with the same letter in each category were not considered significantly different by Tukey’s test. Analysis of variance tables are available in the Appendix A.

Figure 4 shows tensile modulus values for PHB and the composites containing raw and torrefied rice hulls. PHB had a tensile modulus of 1743 MPa. In comparison, composites containing coarse and fine RRHs had tensile moduli of 2328 Mpa and 2376 Mpa, respectively. Additionally, composites containing coarse and fine TRHs had tensile moduli similar to PHB without filler, with values of 1795 MPa and 1717 MPa, respectively. The increase in tensile modulus (or stiffness) of the composites with RRHs could be attributed to the stiffer, stronger raw biomass. In comparison, torrefaction of rice hulls resulted in a more friable, brittle material. Particle size did not affect tensile modulus values. Refs. [54,55] also found increases in tensile modulus values for PHB composites containing raw rice hulls. Ref. [54] attributed this increase to the more rigid rice hulls compared to PHB.

The tensile strengths of the composites are shown in Figure 5. The average tensile strengths of PHB and the two composites with RRHs were between 42.4 and 44.5 MPa. In comparison, the average tensile strengths of composites containing coarse and fine TRH were about 25% less than PHB without filler, with values of 33.1 and 33.5 MPa, respectively. Tensile strength depended on the effective stress transfer between matrix and filler [35]. Typically, composites had lower strength than the plastic without filler due to weak filler, poor adhesion, and poor dispersion.

Figure 6 shows the percent elongation results for PHB and the composites. PHB had an elongation of 6.43 ± 0.32 %, whereas PHB with coarse RRH, fine RRH, coarse TRH, and fine TRH had elongation values of 3.83 ± 0.27%, 3.88 ± 0.35%, 4.13 ± 0.07%, and 5.26 ± 0.36%, respectively. The composite containing fine TRH had a 20–25% greater elongation value than the other composite samples. [55] also found lower elongation at break values for PHB/raw rice hull composites compared to neat PHB, whereas [54] found higher maximum strain at break values.

Figure 7 shows representative flexural stress–strain curves and Figure 8 shows flexural modulus results for PHB and composite samples. PHB had a modulus value of 8633 MPa. In comparison, samples containing RRH showed a 50–58% increase in modulus compared to PHB without filler and samples with torrefied rice hull. This agreed with results from [35], which showed that composite stiffness depended on filler stiffness rather than particle-matrix adhesion. The composites with stiffer, stronger raw rice hulls had higher modulus than composites with the more friable, brittle torrefied rice hulls.

Flexural stress at yield results for PHB and composites are shown in Figure 9. Average stress at yield values for RRH samples exhibited a 4–5% increase when compared to PHB, whereas TRH composite samples exhibited a 20–25% decrease in comparison with PHB. The TRH composites exhibit lower flexural stress at yield than the plastic without filler because they contained weak fillers, had poor adhesion between filler and plastic matrix, and/or had poor dispersion of the fillers within the matrix.

Our results show that RRH can be used as filler at 10% without effecting mechanical strength, while TRH incorporation leads to decreases in tensile and flexural strength. [37] found that the incorporation of 10% raw or torrified wheat straw led to a major decrease (>30%) in stress at break. While our TRH results are similar, our RRH compounds maintained their strength. This could be due to the use of peroxide crosslinking, leading to greater compatibility of the RRH hulls in the PHB matrix. The intrinsic properties of rice hulls, which contain more silica than wheat straw, could have also contributed to our result. In concordance with Berthet et al., we found the use of RRH did lead to a slight stiffening of the material, increasing the modulus and increasing strain at break.

### 3.6. Fourier Transform Infrared (FTIR) Spectroscopy

Lignocellulosics are generally hydrophilic materials due to the presence of oxygen-rich components such as hemicellulose and cellulose. Figure 10 shows the FTIR absorbance spectra for the fillers (raw rice hulls and torrefied rice hulls) and the matrix (PHB). As seen from the spectrograph of the raw rice hulls, three distinct absorbance peak ranges appeared in 700–1200, 1400–1800, and 2800–3800 cm^−1^, corresponding to C-O (hemicellulose, cellulose, and lignin), C-O (hemicellulose and lignin), and -OH (water, cellulose, and hemicellulose), respectively. The most notable change was in the intensity of the peaks in the range of 2800–3800 cm^−1^, which decreased upon torrefaction, probably due to the complete and partial decomposition of the hemicellulose and cellulose [48]. This resulted in a structurally hydrophobic product, as was observed in other studies [29,48,56]. The PHB spectra was dominated by the characteristic carbonyl stretch at 1718 cm^−1^. Compared to the fillers, PHB did not exhibit an absorbance in the range of 2800–3800 cm^−1^. The sharp peaks at 2800–2900 cm^−1^ are identified as C-H from methyl and methylene groups. The increase in oxidation and hydrophobicity may have led to a weakening of interactions between the TRH and PHB, as well as reduced the capacity of the peroxide generated radicals to crosslink RH to the PHB chains.

### 3.7. Thermomechanical Analysis (TMA)

Heat distortion temperature (HDT) is a measure of a composite’s resistance to deflection under a given load and increased temperature. HDT can be used to determine a safe maximum temperature for different applications [36]. Figure 11 shows HDT results for PHB and the composite samples. Coarse RRH had the highest average HDT of 161.9 °C and the composites ranked as follows: coarse RRH > PHB > fine RRH > coarse TRH > fine TRH. The sample containing coarse RRH had a 21.5 °C higher HDT than the sample containing fine TRH. HDT improved with enhanced interfacial adhesion and chemical bonding between filler and plastic matrix [57].

### 3.8. Scanning Electron Microscope

Figure 12 and Figure 13 show images of the cross sections of injection molded samples. The PHB exhibits a somewhat wavy appearance due to the stress of fracturing under liquid nitrogen. Raw rice hull and torrefied rice hull particles had varied shapes, with some having a combination of multiple roughly cubic structures (Figure 13b) and other particles having more variable and complex structure (Figure 12b). The SEM images suggest generally good dispersion of the filler with no indication of particle clustering or agglomerations.

Aside from dispersion pattern, SEM images can also show the interfacial adhesion between the filler and matrix. Image (a) of Figure 12 includes four features with red numbered arrows. Feature 1 shows a shallow hole from a filler particle pulled out during cryo-fracturing. Feature 2 (2nd from left side) shows a deeper hole from a filler particle. Feature 3 (3rd from left) shows a complete filler particle on the surface of the fracture, while feature 4 (right-most) shows another shallow hole left from a filler particle. The presence of holes and whole particles indicated that during cryo-fracturing, particles were either pulled out or left intact, implying that the strength of the particle was greater than the filler-matrix adhesion and that filler-matrix adhesion was poor. Image (b) of Figure 12 is a higher magnification of feature 3 and shows shadows around the edges of the particle (marked by red arrows), indicating poor adhesion between the filler and plastic matrix. Figure 13 shows images of the cryo-fractured RRH composite sample, where edges of filler particles were wetted in plastic without the appearance of shadows, which indicates good particle-matrix adhesion. These images may suggest that the torrefaction treatment of fillers is not necessary in PHB matrices.

## 4. Conclusions

A comparison of TRH and RRH filled PHB composites was made from extruded and injection molded samples. PHB is a renewable, biodegradable alternative to petroleum-based plastics. The development of renewable and biodegradable polymer composites containing natural fibers have the potential to reduce the end-of-life environmental impact of plastic products and improve the utilization of agricultural byproducts. The current research showed that the addition of RRH fillers resulted in light or medium brown colored composites and TRH fillers resulted in dark black composites, demonstrating that RRH and TRHs can be used as renewable-sourced colorants in plastic products, and are an alternative to petroleum-sourced colorants such as carbon black. Mechanical testing showed RRH composites exhibited improved tensile and flexural modulus, comparable tensile strength and flexural stress at yield, but reduced elongation compared with the PHB. TRH composites exhibited comparable tensile and flexural modulus, but weaker tensile strength, reduced elongation, and reduced flexural stress at yield in comparison to PHB. The better mechanical properties of RRH composites in comparison to TRH composites were due to better particle-matrix adhesion. CHNSO analysis showed reduced oxygen concentration in the torrefied rice hulls in comparison to raw feedstock. Additionally, FTIR analysis showed RRH particles contained more surface functional groups containing oxygen than TRH particles, indicating that RRHs should be more compatible with the polar PHB plastic. SEM images showed shadows between filler and plastic in TRH composites and better wetted filler particles in the RRH composites, indicating that polar PHB was more compatible with RRHs.

## Figures and Tables

**Figure 1 polymers-14-03882-f001:**
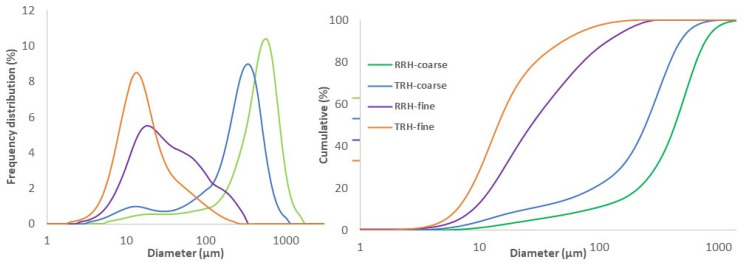
Particle size distribution for raw rice hulls (RRH) and torrefied rice hulls (TRH) shown by frequency distribution (**left**) and cumulative percentage (**right**).

**Figure 2 polymers-14-03882-f002:**
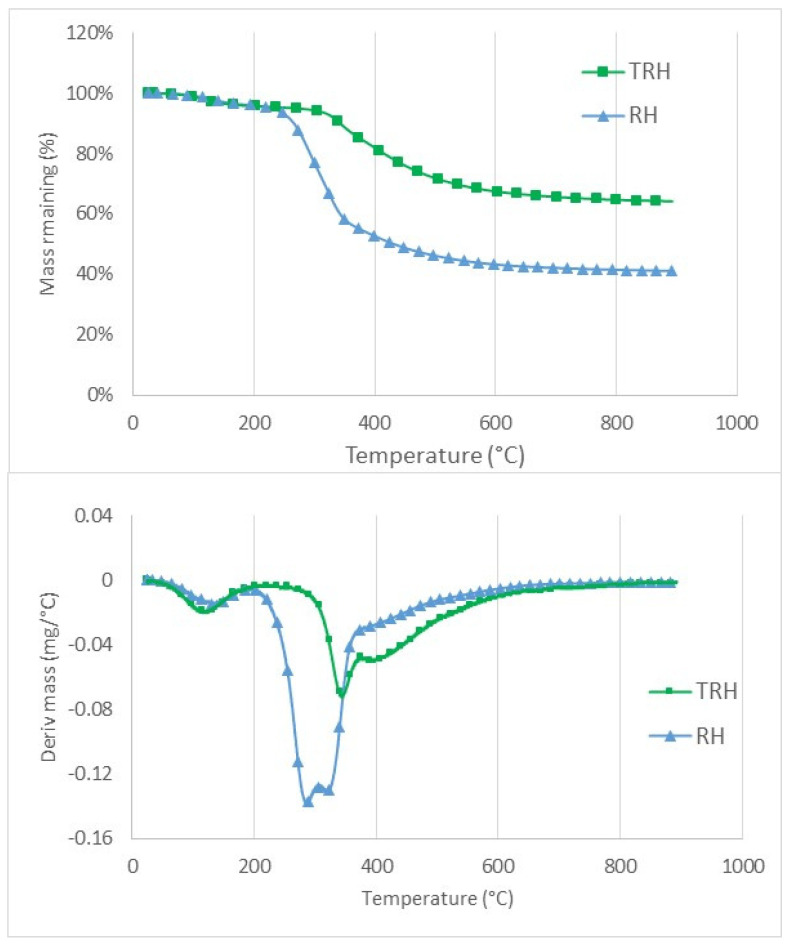
TG (**top**) and DTG (**bottom**) curves for raw rice hulls (RRH) and torrefied rice hulls (TRH).

**Figure 3 polymers-14-03882-f003:**
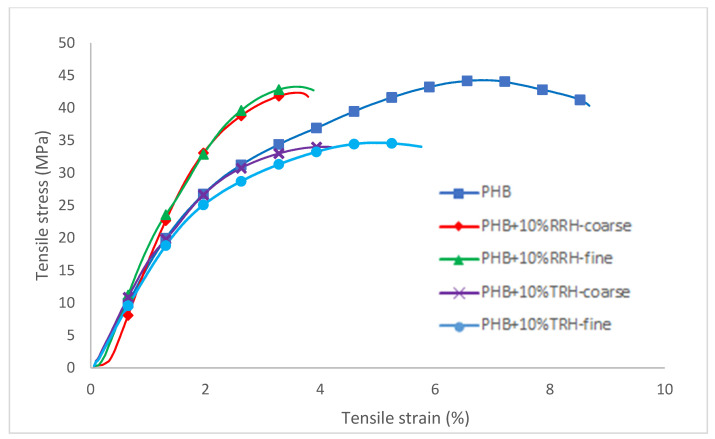
Stress–strain curves for tensile tests of PHB, PHB with 10% coarse RRH, PHB with 10% fine RRH, PHB with 10% coarse TRH, and PHB with 10% fine TRH composite samples.

**Figure 4 polymers-14-03882-f004:**
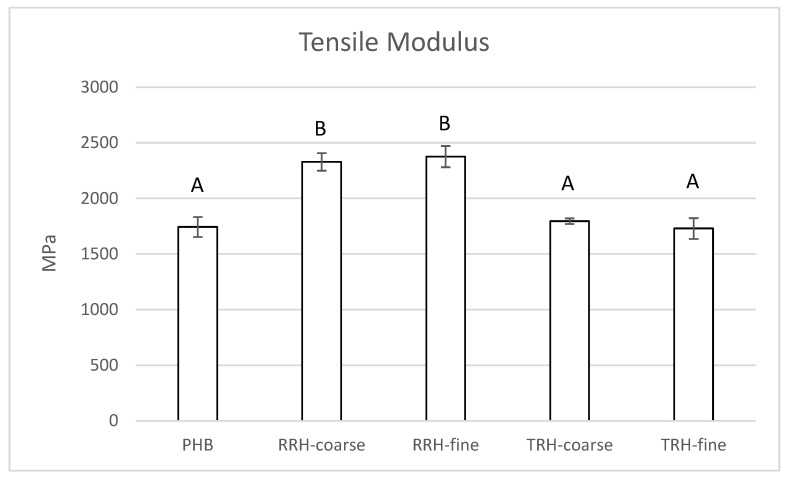
Tensile modulus measured for PHB, PHB with 10% coarse RRH, PHB with 10% fine RRH, PHB with 10% coarse TRH, and PHB with 10% fine TRH composite samples. Means with the same letter in each category are not significantly different by Tukey’s test (α = 0.05).

**Figure 5 polymers-14-03882-f005:**
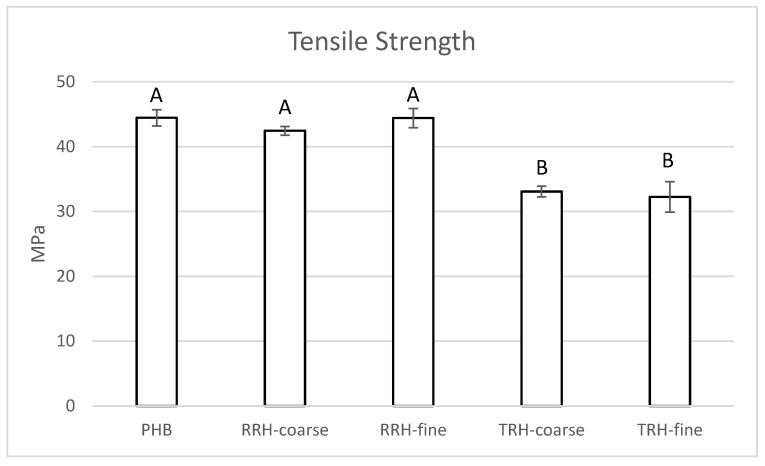
Tensile maximum strength measured for PHB, PHB with 10% coarse RRH, PHB with 10% fine RRH, PHB with 10% coarse TRH, and PHB with 10% fine TRH composite samples. Means with the same letter in each category are not significantly different by Tukey’s test (α = 0.05).

**Figure 6 polymers-14-03882-f006:**
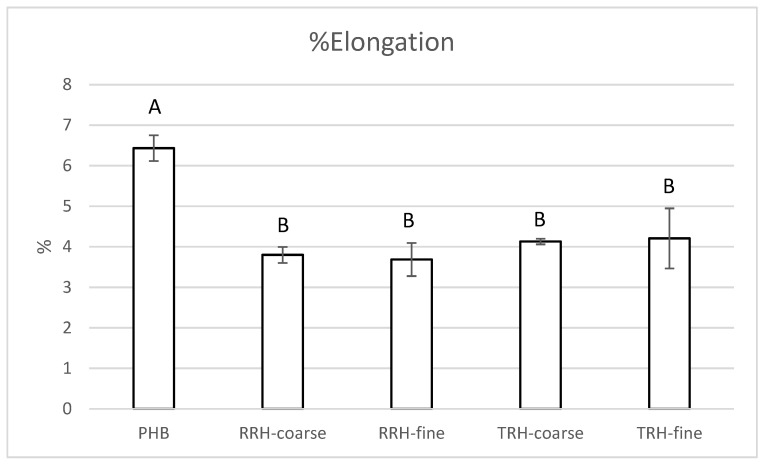
Tensile percent elongation measured for PHB, PHB with 10% coarse RRH, PHB with 10% fine RRH, PHB with 10% coarse TRH, and PHB with 10% fine TRH composite samples. Means with the same letter in each category are not significantly different by Tukey’s test (α = 0.05).

**Figure 7 polymers-14-03882-f007:**
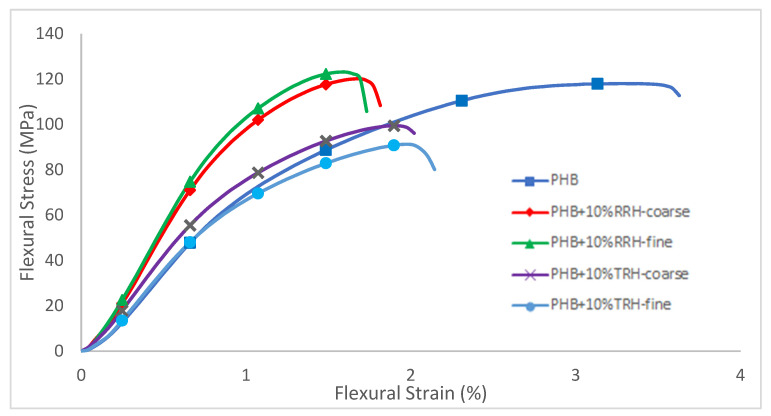
Stress–strain curves for flexural tests of PHB, PHB with 10% coarse RRH, PHB with 10% fine RRH, PHB with 10% coarse TRH, and PHB with 10% fine TRH composite samples.

**Figure 8 polymers-14-03882-f008:**
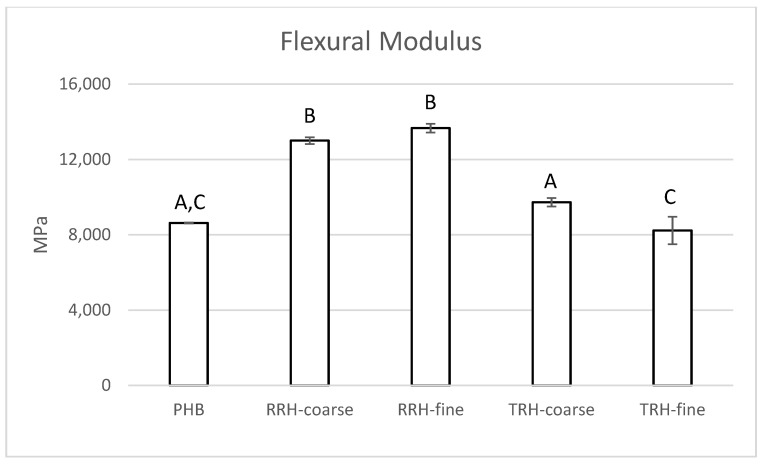
Flexural modulus measured for PHB, PHB with 10% coarse RRH, PHB with 10% fine RRH, PHB with 10% coarse TRH, and PHB with 10% fine TRH composite samples. Means with the same letter in each category are not significantly different by Tukey’s test (α = 0.05).

**Figure 9 polymers-14-03882-f009:**
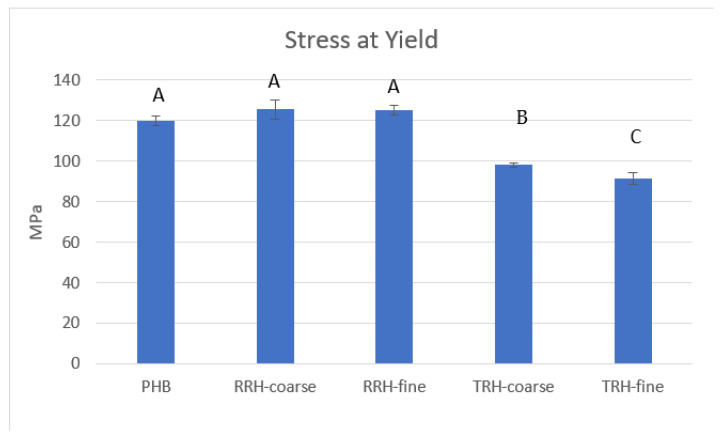
Stress at yield measured for PHB, PHB with 10% coarse RRH, PHB with 10% fine RRH, PHB with 10% coarse TRH, and PHB with 10% fine TRH composite samples. Means with the same letter in each category are not significantly different by Tukey’s test (α = 0.05).

**Figure 10 polymers-14-03882-f010:**
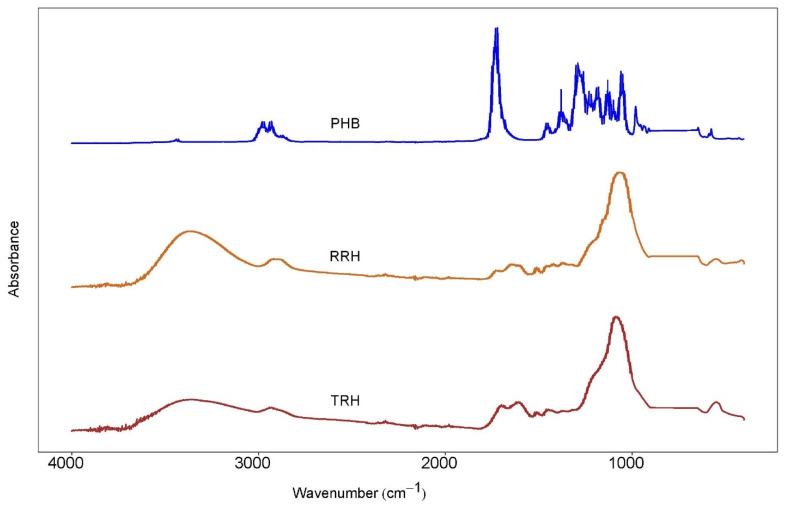
FTIR absorbance spectra of (top) PHB, (middle) raw rice hulls (RRH), and (bottom) torrefied rice hulls (TRH).

**Figure 11 polymers-14-03882-f011:**
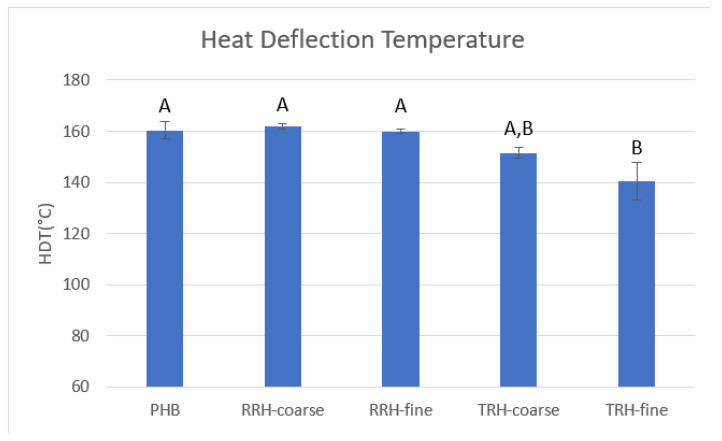
Heat deflection temperatures measured using TMA for PHB and PHB composites. Standard deviation of measurements listed in parentheses. Means with same letter are not significantly different by Tukey’s test (α = 0.05).

**Figure 12 polymers-14-03882-f012:**
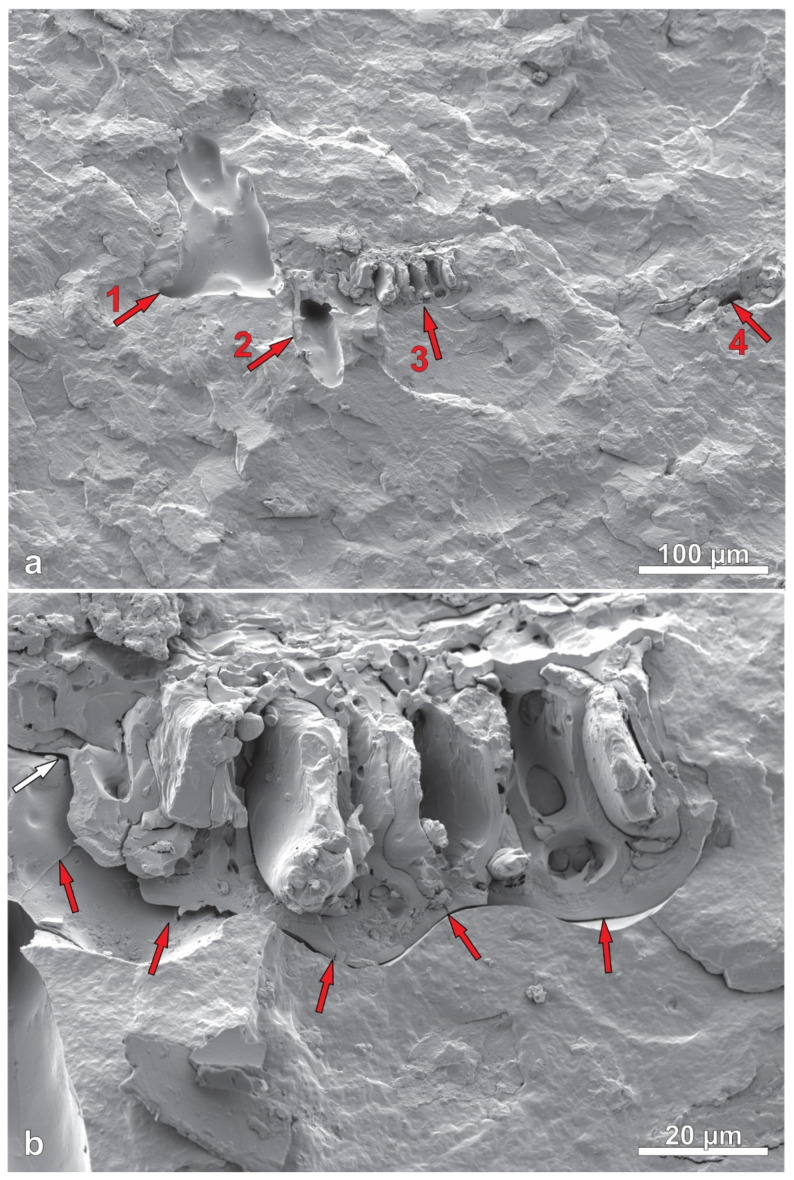
Images of cryo-fractured TRH composite. Image (**a**) has 4 numbered arrows (in red): (1) shallow hole left from TRH particle, (2) deeper hole left from TRH particle, (3) TRH particle, and (4) shallow hole from TRH particle. Red arrows on image (**b**) shows space between particle and matrix, indicating poor adhesion.

**Figure 13 polymers-14-03882-f013:**
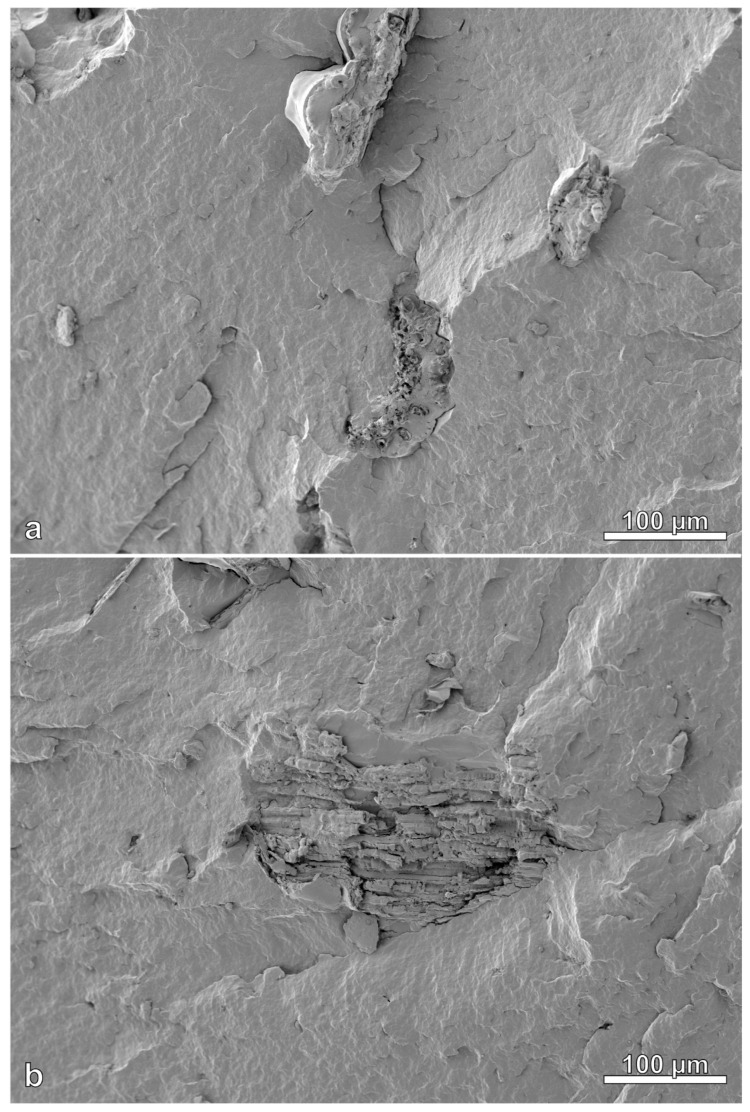
Two images (**a**,**b**) of cryo-fractured RRH composite show good adhesion between filler particles and plastic.

**Table 1 polymers-14-03882-t001:** CHNSO elemental and ash composition of raw rice hulls (RH) and torrefied rice hulls (TRH). Standard deviation is listed within parentheses. Ash content was calculated by difference.

	C [%]	H [%]	N [%]	S [%]	O [%]	Ash [%]	Total [%]
RH	38.70 (0.14)	4.92 (0.01)	0.47 (0.01)	0.03 (0.01)	37.15 (0.28)	18.73 (0.41)	100.00
TRH	42.60 (0.26)	4.33 (0.04)	0.57 (0.01)	0.02 (0.00)	17.34 (0.17)	35.14 (0.47)	100.00

**Table 2 polymers-14-03882-t002:** Percent crystallinity for the extruded composites.

	Crystallinity [%]
PHB	54.5%
PHB+10% RRH coarse	54.3%
PHB+10% RRH fine	54.8%
PHB+10% TRH coarse	55.0%
PHB+10% TRH-fine	54.2%

## Data Availability

Data available on request.

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
