# Peer review of "Polyhydroxybutyrate Rice Hull and Torrefied Rice Hull Biocomposites"

_polymers, 2022, doi:10.3390/polym14183882_

Round 1

Reviewer 1 Report

The manuscript by Dr. McCaffrey and co-authors presents an interesting modification of P(3HB) and its characterization. Although rationale is good,  the results indicate that the chosen filler materials led to minor increase on P(3HB) properties, if any.

General comments:

a) The references are really new, but some of the original and classic references on PHA are missing. Please consider combining the original literature with 2021, 2022 manuscripts. For example: Lee, S. Y. (1996). Bacterial polyhydroxyalkanoates. Biotechnol. Bioeng. 49, 1–14.; Madison LL, Huisman GW. Metabolic engineering of poly(3-hydroxyalkanoates): from DNA to plastic. Microbiol Mol Biol Rev. 1999 Mar;63(1):21-53. doi: 10.1128/MMBR.63.1.21-53.1999. PMID: 10066830; PMCID: PMC98956;

b) All the figures: please enhance quality and work on more elaborate graphics.

Specific comments

1) L 48-49. 'Poly(3-hydroxybutyrate)' is the correct nomenclature.

2) L 51-52. PHA can also be produced by different Archaea. Please check doi: 10.1016/j.nbt.2016.05.001

3) L 54. "Fermentation" should not be used as a synonym for bacterial cultivation, but only for a fermentative bioprocess. Please check if this is the case and clarify if necessary.

4) L 59-62. P(3HB) homopolymer is brittle, but this is one of the may types of PHA. There are many industrially relevant members of the PHA family (copolymers, terpolymers) with a range of thermo-mechanical characteristics (10.1016/j.ijbiomac.2022.06.024 ). Please check and cite the appropriate references on the Handbook of Polyhydroxyalkanoates https://www.taylorfrancis.com/books/edit/10.1201/9780429296635/handbook-polyhydroxyalkanoates-martin-koller

5) L 65-66. Are rice hulls used as food for animals? Would it compete with food supply (human or animals)?

6) L 76. Please correct the first word of the sentence.

7) L 148. Standardize your units, in this case to J/g or J.g-1

8) Figure 1. Do you have any hypothesis regarding the non-single peak on RRH-fine sample?

9) Figure 2. Figure quality is really low. Correct temperature unit to °C.

10) L 328-329. The main goal of the research was to improve the P(3HB) physical properties (L 62-64). Please clarify if the composites presented  improved, null or non-improved physical properties when compared to P(3HB). Compare your results with different PHA copolymers, such as P(3HB-co-3HHx), P(3HB-co-4HB) to improve the discussion section.

11) L 332. 'strength'

12) L 366. Please elaborate and improve the discussion on the SEM results.

13) L 395-397. Elaborate on the potential advantages of producing colored P(3HB), what are the potential applications and advantages compared to commercial polymers?

Reviewer 2 Report

This paper is a significant study of the Polyhydroxybutyrate Rice Hull and Torrefied Rice Hull Biocomposites using extrusion and injection molding. Experimental results are good. However there are some concerns and clarifications need authors attentions are necessary, as:

The authors need to highlight the novelty of the work presented. Please, demonstrate more obvious your innovation in this work. Would you mind identifying blatant discrimination between your work and others?

Please, redraw the Figure 2 because it is not clear (x, y graduation and axis titles) and indicate the different peaks. Also, indicate the important peaks in the Figure 10.

Table of ANOVA Analysis is missing.

Please, it is necessary that the authors present and more discussed the results of FTIR and compare with other work in the literature.

A complete revision of the document is necessary. Improved bibliography, there are more recent references not cited.

Round 2

Reviewer 1 Report

The authors replied to all the comments and improved the text accordingly. It is now suitable for publication on Polymers.

Reviewer 2 Report

In my opinion, the author has revised and improved the thesis according to the requirements, so I suggest to accepting it.